# The Effect of Hard Pecking Enrichment during Rear on Feather Cover, Feather Pecking Behaviour and Beak Length in Beak-Trimmed and Intact-Beak Laying Hen Pullets

**DOI:** 10.3390/ani12060674

**Published:** 2022-03-08

**Authors:** Paula Elizabeth Baker, Christine Janet Nicol, Claire Alexandra Weeks

**Affiliations:** 1Laying Hen Welfare Forum, 22 City Road, London EC1Y 2AJ, UK; 2Royal Veterinary College, Hawkshead Lane, Brookmans Park, Hatfield AL9 7TA, UK; cnicol@rvc.ac.uk; 3Bristol Veterinary School, University of Bristol, Langford House, Bristol BS40 5DU, UK; claire.weeks@bristol.ac.uk

**Keywords:** rear, injurious pecking, pullet, pecking pan, beak length

## Abstract

**Simple Summary:**

Laying hens may peck each other, painfully pulling out feathers and damaging flesh. There are many potential reasons and no obvious solutions, so the sharp tip of the beak is often carefully removed after hatching to limit the damage inflicted. Scientist have found it is useful to provide hens with lots of other things to forage in and peck at. Our study compared the effect of providing pans containing an abrasive material designed to attract pecking behaviour to eight treatment flocks of young hens, with eight flocks acting as controls. Each group had four beak-trimmed and four intact-beak flocks paired by farm. The pans significantly reduced the side and top beak lengths in samples of birds measured at 6–7 weeks and 10–11 weeks of age, and beak growth appeared to be linear. Thus, provision of abrasive material in pans effectively blunted beaks of younger birds. By the end of rear (14–15 weeks), birds seemed to lose interest in pecking at the pans, so there was no difference between treatment and control flocks, indicating scope for redesign to retain their interest.

**Abstract:**

Injurious pecking, commonly controlled by beak trimming (BT), is a widespread issue in laying hens associated with thwarted foraging. This controlled study compared the effect in intact and beak-trimmed pullets of providing pecking pans to eight treatment flocks from six weeks of age. Flocks (mean size 6843) comprised eight British Blacktail, six Lohmann Brown and two Bovans Brown. All young birds (6–7 weeks) pecked more frequently at the pecking pans (mean 40.4) than older pullets (mean 26.0, 23.3 pecks/bird/min at 10–11 weeks and 14–15 weeks, respectively) (*p* < 0.005). There was no effect on feather pecking or plumage cover. Mean side-beak length and mean top-beak lengths were shorter in treatment flocks at 6–7 weeks and 10–11 weeks (*p* < 0.001). Intact-beak treatment flocks had shorter mean side-beak length at 10–11 weeks (*p* < 0.001) and at 14–15 weeks (*p* < 0.05) and mean top-beak length at 6–7 weeks (*p* < 0.05) and at 10–11 weeks (*p* < 0.05). BT treatment flocks had shorter side-beak and top-beak lengths at 6–7 weeks and at 10–11 weeks (*p* < 0.001). Beak lengths showed linear growth, with individual bird variation indicating a potential for genetic selection. The study demonstrated that abrasive material can reduce beak length in pullets.

## 1. Introduction

Injurious pecking (IP) is an umbrella term for inter-bird pecking behaviours including severe feather pecking (SFP), vent pecking (VP), cannibalistic pecking (CP) and toe pecking (TP) [1]. The forceful pulling of feathers of conspecifics that occurs during SFP not only causes pain and distress [2] but can lead to substantial feather loss and be associated with cannibalistic behaviour, vent pecking and a decline in egg production [1,2] as well as disease and high mortality [2,3]. Birds with poor feather cover due to IP may consume up to 40% more feed [4]. The risk of IP is multifactorial, with the environment, nutrition and genetics all having a role [5,6], making IP hard to manage in commercial flocks [5,7].

A further form of inter-bird pecking is gentle feather pecking (GFP), which comprises repeated pecks at the edges of feathers—often aimed at the tip of the tail and sometimes other areas. GFP results in minor damage of little welfare concern [5]. The occurrence of GFP in pullets is not a good predictor for subsequent SFP or other forms of IP in older birds, but there can be a positive association between GFP and SFP in birds of the same age [8]. Both GFP and SFP behaviours can be witnessed during the rearing period. GFP has been reported in week old chicks [9] and can peak around three to four weeks of age [10]. The first signs of SFP in young pullets have been observed as pecking around the preen gland and oily feathers near the tail [11]. SFP has been observed in flocks at 10 weeks of age [12].

Beak trimming (BT) remains one of the most effective management tools to reduce the impact of IP [1,13,14], as blunted beaks cause less damage and pain to the recipient bird. Plumage damage is also reduced, as beak trimmed birds tend to perform less destructive feather pecking [15,16]. Issues related to BT remain controversial worldwide and still provoke a great deal of debate from an animal welfare perspective [1,17].

There are two methods: the hot blade technique (HB), still widely used in some countries, and infra-red beak trimming (IRBT), which has replaced HB trimming of day-old chicks in the UK and in Europe [18,19].

Hot-blade trimming involves the removal of up to one third of the maxilla (upper) and (lower) mandible [1,17], which can include the horny skin structure of keratinized epidermis [11,18]. This technique can result in short and long-term pain, causing tissue damage, nerve injury and the development of neuromas and pain-related behaviours when trimmed at five weeks of age [20,21], as well as reduced feed intake and a decline in growth rate [22].

IRBT is recognised as less damaging with fewer adverse consequences, suggesting a low risk of long-term pain [23]. However, IRBT is associated with short-term reductions in feeding and ground pecking behaviour and reduced weight gain observed in the first few days [21,24,25]. Although these effects resolve by the time birds are two to four weeks of age [26,27], there is some adverse impact of IRBT on birds. Thus, effective management of IP behaviour would obviate the need to perform BT.

There is increasing awareness of the impact of the environment and management during rear on the development of IP in the subsequent laying period [2,3]. Young birds kept in barren environments with limited enrichment and foraging opportunities may exhibit fear, reduced cognitive ability and be attracted to peck at feathers [28,29,30]. In contrast, the provision of high-quality litter and other enrichments from an early age can direct foraging and exploratory behaviour towards harmless substrates and has been shown to reduce subsequent overall feather pecking [2], SFP [31] and VP [32] on the laying farm. Despite the influence of the rearing period, adult hen behaviour remains responsive to the environment, and the continued provision of appropriate foraging opportunities for adult birds is important [33]. A key approach to reducing the risk of IP in adult birds is to provide appropriate substrates with the aim of satisfying foraging motivation [34]. Several other management strategies, for example those that address lighting, diet or use of the range, can also be effective. In recognition of this complexity, the effectiveness of management packages that combine a range of strategies in a holistic attempt to prevent or reduce injurious pecking (IP) has been assessed [31,35]. One study of commercial free-range laying hens provided 53 treatment flocks with a bespoke management package and compared their behaviour and performance with 47 control flocks where no changes were suggested [31]. IP and plumage damage were notably lower within the treatment flocks and less SFP was observed. All types of feather pecking were reduced by increasing the number of management strategies in place [31]. A further study found that the concurrent provision of a variety of objects designed to attract pecking behaviour and the provision of outside shelters to encourage range use resulted in a reduction in both GFP and SFP [35].

A complementary approach with a different focus is to try to maintain birds with “naturally” blunted beaks, which may be achieved by providing hard materials as pecking substrates. Commercially available durable pecking blocks and stones have two potential functions, first, by reducing or blunting beaks to avoid beak trimming and, second, by providing a type of enrichment. Some studies have examined the effects of providing hard materials as part of a package of changes. For example, Pettersson et al. [35] included pecking pans as part of a resource package on 14 laying farms and considered them well-used in most flocks. As stated earlier, the package as a whole was effective in reducing feather pecking behaviour, but the separate contribution of the pecking pans could not be evaluated. In addition, due to faecal contamination and perching activity, it was found that pecking bouts significantly decreased over time [35]. When foraging and pecking enrichments such as pecking stones, pecking blocks and lucerne bales were combined with a reduction in stocking density (SD) in laying hen chicks from 22.9 birds/m^2^ to 18.1 birds/m^2^, a significant reduction in GFP and SFP was seen. On average, the highest rate of enrichment pecking per bird per 3 min was observed for lucerne bales, followed by pecking stones and pecking blocks, but, again, the separate effect of the pecking stones could not be evaluated [16,32].

Whilst the studies mentioned above have included hard pecking materials, they have not examined whether these materials are effective in blunting birds’ beaks via abrasion. Abrasion could be an alternative to BT in laying hens, but this has only been examined in a handful of studies. Fiks-van Niekerk and Elson [36] placed an abrasive material inside a feeding trough at rear (from 6 weeks) and chain feeder troughs at lay (from 18 weeks). Birds’ beaks were measured at four-week intervals with vernier callipers for length and curvature by tape measure. The continuous exposure to the abrasive materials throughout rear and lay was effective and resulted in consistent shortening of the beaks by an average of 1 mm. However, the abrasive material was less effective when provided only during the rearing period, and it was difficult to standardise the measurement of the beak tip (hook) on immature birds. Iqbal and others [1] likewise explored the effect of pecking stones on beak length, plumage condition, toe length, production and mortality from 16 weeks until 66 weeks of age. Morrissey et al. [30] investigated the effect of beak blunting boards as part of a package of other enrichments but found no effect on the beak length or beak sharpness on the upper maxilla. Further research [37] in a replicated smaller study with beak blunting boards and cuttlebone found that at 29, 35 and 40 weeks of age, hens pecked at a cuttlebone more than beak blunting boards, performed less feather pecking behaviour and had less plumage damage. In addition, there was a suggestion that cuttlebone might reduce beak length (*p* = 0.070), but there were no effects on other beak measures and no treatment effects of enrichment or location on any measure. However, pecks at the extra enrichment generally decreased over time. In other studies, pecking stones have been more successful as beak blunting tools; for example, in turkeys after six months of access to pecking stones, the beaks of about 80% of treatment birds resembled those of beak trimmed conspecifics [38].

In recent years, there has been a shift in the technique used to measure hen beak characteristics. Previous work in other avian species has revealed diversity in beak length, depth and strength in Darwin’s finches [39] using landmark-based geometric morphometrics. This methodology has more recently also been used for chickens and turkeys [30,37,40,41]. Although precise, with reference points identified on the upper beak, lower beak, overhang and angle of the beak tip, the method is very time consuming and was not feasible for the current project conducted on a commercial farm. Thus, studies have generally focused on enrichment preference in older laying hens and have been undertaken in small group in-house experiments [30,37], with few studies conducted on young pullets at different age ranges. The potentially differential effects of hard pecking materials on intact-beak and beak-trimmed birds in a commercial setting have not been considered. The aim of our current study was therefore to conduct a controlled investigation in commercial flocks of the effects of a hard pecking material on the behaviour and beak characteristics of beak-trimmed and intact-beak laying hen genotypes during rear (from 6 to 15 weeks of age).

## 2. Materials and Methods

### 2.1. Animals and Housing

Sixteen UK commercial rearing flocks were studied between September 2015 and December 2016. Half of the 16 flocks were infra-red beak trimmed (IRBT) at day old at the hatchery and the other half were intact-beak flocks (i.e., not trimmed). Strains were British Blacktail (BBT) (*n* = 8), Lohmann Brown (LB) (*n* = 6) and Bovans Brown (BB) (*n* = 2). Flock size ranged from 3300 to 11,000 and the average flock size was 6843 (see Table 1). All flocks were reared in loose housing systems with wood shavings or shredded paper as the floor substrate. Most of the flocks had A-frame perches, except for two, which had a raised slatted system (flocks 5 and 6). All flocks had chain feeders. Four flocks were reared on organic mash feed and the others were reared on standard commercial mash. No flocks had access to the outside. Welfare and behaviour assessments took place over three visits: 6–7 weeks of age, 10–11 weeks and 14–15 weeks of age. Eight treatment flocks were supplied with pecking pans (Vencomatic UK) and eight were paired control flocks reared on the same farm Flocks were allocated to pecking pan or no pecking pan treatment groups based on matching strains and beak status (and reared concurrently as sister flocks on each farm). The experimental design consisted of four intact treatment, four intact control, four trimmed treatment and four trimmed control flocks.

Pecking pans (Figure 1) comprised a green plastic pan feeder with a detachable grey base. The pan contained a hard block composed principally of sand, cement and oyster shell. Each treatment flock was given one pecking pan per 500 birds, and these evenly distributed on the litter area throughout the rearing house. Pecking pans were placed the day before the first observations commenced, at around 6–7 weeks, to allow the birds to acclimatise to them, and they remained for the rest of the rearing period.

### 2.2. Behavioural and Welfare Assessments

#### 2.2.1. Pecking Pan Interaction

Focal sampling techniques were used to record usage of the pecking pan in treatment flocks. To encourage the birds to make an initial approach to the pans, two handfuls of organic mixed corn were sprinkled into each pan the afternoon before data collection took place on the first visit at 6–7 weeks of age. Observations were made on randomly selected pecking pans and the quadrat around the pan was an estimated 1 m^2^ area. A 2-min habitation period allowed the birds to become accustomed to the presence of the observer, who was sited unobtrusively some distance away. The number of pecks to the pecking pan was observed for two individual birds, 2 min per bird, 4 min in total. If either of these birds discontinued pecking and left the area, the observation ended, and its time was recorded.

#### 2.2.2. Use of Pecking Pans

An initial aim was to record use of the pecking pans at each age of 6–7, 10–11 and 14–15 weeks and to classify usage as minimum, medium and maximum wear, with photographs taken to aid evaluation. However, it proved difficult to assess wear and compare or weigh the pecking substrate throughout the three visits. This was due to the substrate not being standardised in each pecking pan and because, from time to time, the pecking pan itself would contain bird excrement. Thus, this part of the trial was abandoned, and data not used.

#### 2.2.3. Feather Pecking Behaviour

Observations of feather pecking behaviour were recorded at each age (6–7, 10–11 and 14–15 weeks) between the hours of 08.30 a.m. and 12.00 noon and generally matched for time of day across all visits. Five-minute observations of behaviour were recorded in 2 m^2^ quadrats in nine randomly selected areas of the house (total of 45 min) to give a representative sample. A 2-min habituation period allowing the birds to become accustomed to the presence of the observer was used and observations were taken at a 1 m distance to minimise bird disturbance. The number of birds in the observation area was counted before and after the observation to calculate the average number present during each observation period. Observations included all bouts of gentle feather pecking (GFP), where a bout was defined as continuous pecking until another behaviour was performed or the behaviour stopped for a 5 s gap [42]. All individual instances of severe injurious pecking (SFP), vent pecking (VP), cannibalistic pecking (CB) and aggressive pecking (AP) were also recorded (see Table 2 for definitions).

### 2.3. Feather Scoring and Plumage Condition Assessment

Plumage condition was assessed at the end of the rearing period (bird age at 14/15 weeks). This time point was selected as young chicks and pullets go through many moults throughout rear. Twelve birds per area (108 birds in total) were visually scored during the morning behaviour observations in nine areas of the house on the litter area. Birds were not handled in order to minimise flock disturbance. Pullets were randomly selected by counting three birds to the right of the first bird focused upon. Feather cover was assessed and scored for five areas of the body—neck, back, tail, rump and wings—using a five-point scale and photographs for reference, a method similar to Bright and others (2006) (0 = No damage to 4 = Severe damage to skin and very large, injured areas (>5 cm^2^ traumatized)). Birds that were selected for beak measurements (45 in total) were further inspected for plumage condition using the same five-point scale (Table 3).

### 2.4. Measurement of Beak Length

A sample of the birds (30–45 per visit at 6–7 weeks, 10–11 weeks and 14–15 weeks) was selected from different areas of the rearing house for beak measurement. Birds were randomly selected using the method described above. Birds were caught in the afternoon so the flock behaviour during observations would not be influenced by catching and handling of the sampled birds. Birds were gently restrained by one person and wrapped in a disposable cloth to restrain their wings so they could be easily handled. The top of the right leg of each bird was marked with a black dot with a permanent marker pen to ensure that these birds would not be recaptured throughout the study. Beak length was measured with vernier callipers using methodology slightly adjusted from that of Van de Weerd (2005) [36] (see Table 4 and Figure 2 and Figure 3). During the study, it became apparent that it was difficult to take accurate hook measurements and to standardise the technique, so these data were not used. Each measurement was repeated three times in succession to provide a mean length with an estimated accuracy of ±0.5 mm.

### 2.5. Statistical Analysis

The data collected from the study were evaluated for significant relationships using statistical tests in IBM SPSS version 26. General linear modelling (GLM) was used, with boot-strapping procedures where appropriate to account for variable data distributions, to examine effects of treatment (pecking pans provided or not), beak status (intact or trimmed) and bird age at (6–7 weeks, 10–11 weeks and 14–15 weeks), treated as a repeated measure. A flock was taken to be the independent unit for analysis for measures of bird pecking behaviour towards pans or other birds and for plumage condition, as these measures would be highly influenced by the behaviour of other birds in each flock. Pecking activity directed towards pecking pans was analysed for treatment flocks with the model including beak status and bird age. Bird pecking behaviour at the pans was corrected for the length of time for each focal bird and is presented as a pecking rate (pecks/bird/min).

GLM repeated measures analyses were used for analysis of feather pecking (SFP and GFP) in both treatment and control flocks, with models including beak status and bird age. For each flock, a mean value of the nine observations of feather pecking behaviour was calculated for each visit: visit 1 at 6–7 weeks, visit 2 at 10–11 weeks and visit 3 at 14–15 weeks. Plumage condition on non-handled birds was assessed only at the end of the rearing period, and so a simple model examined the effect of treatment and beak status. Comparable simple statistical tests were performed on the 45 birds/flock which were plumage scored during handling for beak measurements.

Analysis of bird beak characteristics was performed at the individual bird level using data for side length and top of beak for each of the 45 birds sampled at each age (6–7 weeks, 10–11 weeks and 14–15 weeks). Normality tests showed that none of the data were normally distributed even after transformation, therefore, non-parametric statistical analyses were used. Mann–Whitney U tests were used to compare side-beak length and top-beak length between intact and IRBT birds across age (6–7 weeks, 10–11 weeks and 14–15 weeks), and patterns of beak growth were plotted for each breed. Further Mann–Whitney U tests were conducted to explore the effect of pecking pan provision on side-beak and top-beak length. A Bonferroni adjustment was used to reduce the possibility of incorrectly rejecting the null hypothesis (Type 1 error) because independent tests were being performed simultaneously on a single dataset (15 tests 0.05/15 = 0.003).

## 3. Results

### 3.1. The Effects of Age on Pecking Pan Use in Intact and Beak Trimmed Flocks

There was a reduction in pecking pan activity (peck rate) with age for the eight flocks with access to the pecking pans, with young birds at 6–7 weeks pecking more often (mean 40.4) compared to 10–11 weeks (mean 26.0) and 14–15 weeks (mean 23.32), pecks/bird/min: F = 25.51; df = 1,6; *p* = 0.003 (Figure 4). However, there was no significant difference between intact or beak trimmed flocks (F = 1.44; df = 1,6; *p* = 0.27).

### 3.2. The Effect of Pecking Pan Presence on Feather Pecking Behaviour

No significant treatment differences in scan observations of SFP were found in birds with or without access to the pecking pans, nor were there beak status effects. For SFP, overall mean pecks/5 min scan were 0.241 (0.016–0.465) with the provision of pans versus a mean of 0.526 (0.285–0.767) with no pans (F = 3.57; df = 1,12; *p* = 0.08). SFP appeared to increase linearly with bird age (F = 7.77; df = 1,12; *p* = 0.016). There were no age or treatment effects on GFP behaviour. The frequency of GFP was 2.58 (se 0.52) for beak trimmed birds and 1.12 (se 0.48) for intact beak birds (F = 4.21; df = 1,12; *p* = 0.06).

### 3.3. The Effect of Pecking Pan Presence in Intact and Beak Trimmed Flocks on Plumage Condition

Visual scoring of plumage condition at 14–15 weeks of age showed no significant differences across all body regions between intact flocks and BT flocks with and without a pecking pan. There was little plumage damage, and none seen in the wings or rump region (Table 5).

Analysis of the flock averages from detailed feather scoring of handled birds confirmed no significant effect of beak status or pecking pan provision on plumage in any of the body regions (Table 6). Again, there was no plumage damage to the rump region.

### 3.4. The Effects of Beak Status and Age on Beak Length

Unsurprisingly, intact beaks were significantly longer than trimmed beaks across all ages. This was seen in both mean side-beak length and mean top-beak length *p* < 0.001) (Table 7 and Table 8). The pattern of beak length increases with age, irrespective of pecking pan provision, and can be seen for each strain in (Figure 5 and Figure 6). A linear rate of increase in mean side-beak length and mean top-beak length was seen in BBT and LB birds, but a growth curve provided a better description of the rate of increase for IRBT BB birds (Figure 5 and Figure 6). There were no intact beak BB flocks in the study.

### 3.5. The Effect of Pecking Pan Presence on Beak Length

When individual birds from all treatment flocks were considered together, irrespective of beak status, mean side-beak length and mean top-beak length were significantly shorter with a pecking pan present at 6–7 weeks and 10–11 weeks (Table 9 and Table 10). No significant differences were found for beak measurements at 14–15 weeks (Table 9 and Table 10).

Intact and IRBT birds were next considered separately. For intact birds, there was a significant shortening effect of pecking pans on mean side-beak length at 10–11 weeks (U = 10,859; *p* < 0.001) and at 14–15 weeks (U = 12,301; *p* = 0.006) and on mean top-beak length at 6–7 weeks (U = 16,414; *p* = 0.026) and at 10–11 weeks (U = 12,202; *p* = 0.004). For IRBT birds, there was a significant shortening effect of pecking pans on mean side-beak length at 6–7 weeks (U = 6378; *p* < 0.001) and at 10–11 weeks (U = 10,819; *p* < 0.001), with similar effects on mean top-beak length at 6–7 weeks (U = 6800; *p* < 0.001) and 10–11 weeks (U = 10,840; *p* < 0.001).

## 4. Discussion

Our study, in common with others [35,37], found that the use of pecking material declined with age. It could be that the pecking pans needed to be placed at an earlier age, at a day old [29], as enrichment can retain interest up until at least 17 weeks of age.

However, we considered that placing the pecking pans early in rear could be detrimental, as the beaks of chicks are soft, and we had concerns that the hardness of the substrate inside the pecking pan could potentially damage the chicks’ beaks. The same pecking pan used in our study was also used in laying flocks [35]. These authors recorded that the pans were used by the birds for the first 15 weeks of lay, but the mean bout duration of pecking at the substrate of 4 s at 25 and 40 weeks decreased over time. Informal observations in our study noted that as the birds matured, the pecking pan was frequently used as a perch rather than a pecking enrichment. The loss of interest in both studies could also be attributed to other factors, such as conspecifics blocking access to the pecking material within the pans and to faecal contamination, indicating the need for refinement of presentation of pecking material. In our study, the maintenance of the pecking pans was the sole responsibility of the farmer. The upkeep varied between farms, with excrement usually not removed, possibly because farmers did not regard the pans as feeding equipment.

Other studies suggest that young chicks and pullets prefer enrichments that they can play with and manipulate, such as string [29]. It is possible that in our study the young birds found the pecking substrate too hard to satisfy their needs to express foraging behaviour. It is also possible that beak-treated birds were more reluctant to peck at something hard owing to short- or long-term pain [22]. Research has shown that feeding habits are different and feed intake is reduced in beak-treated birds [17,22].

We examined whether IP and plumage condition differed between the three strains, finding no significance differences for all areas scored (neck, back, tail and wings). However, relatively low levels of IP were observed and, in general, the flocks had good plumage cover and quality.

It was hypothesized that treatment flocks provided with pecking pans would show reduced levels of feather pecking, yet no differences were observed, and neither were there any significant effects of beak treatment, strain or age. This finding contrasts with other studies, which show that providing some form of enrichment reduces IP during the rearing and laying period [2,31,32]. However, in our study, overall rates of feather pecking were low, many of the sheds had good friable litter and some of the sheds were stocked with a greater space allowance, providing more room for the birds to perform foraging activity, ground scratching and dust bathing behaviour. Most flocks were given the opportunity to pursue priority behaviours (i.e., foraging, perching and dustbathing [42,43,44], perhaps to a greater extent than in some other studies. Supporting this argument are results from works that show that providing enrichments in combination with lower SD can reduce both gentle and severe feather pecking behaviour [7,16,32]. We found no difference in the plumage condition of intact and beak-treated flocks. This unexpected finding is in contrast to other reports which have found that beak trimmed birds have better plumage condition at rear [2,15,44], despite IP activity [45]. This could be explained by the low rate of IP observed in our study, which is a positive finding, as young pullets commonly start to peck around the preen gland and oily feathers near the tail before they are transferred to the laying house [11]. One reason for the relative lack of effects of treatment, strain or beak status on IP may be that short periods of observation at a given time of day were not sufficient to accurately reflect levels of IP, and the sample size was too small to detect differences because this analysis treated whole flocks as independent units in the analysis. Plumage damage is also not solely related to IP and may also reflect moulting [46] or abrasion by furnishings in the shed [38].

Few studies have considered whether pecking enrichments that supply hard substrates can reduce beak sharpness in laying hens, with only three previous studies on birds during the rearing stage. Our study measured beaks at the individual bird level and confirmed, as expected, intact beaks were longer than trimmed beaks. Beak length generally increased with age, except in BB birds, where growth between age 6 and 7 weeks and 10 and 11 weeks was relatively low, followed by a greater increase between 10 and 11 weeks and 14 and 15 weeks of age. Variation between individual birds in beak length tended to decrease with age. Pecking pans had a significant shortening effect on side-beak length and top-beak length at 6–7 weeks of age and at 10–11 weeks. Finding significantly shorter beaks at 6–7 weeks was unexpected, as the pans had been placed just the day before observations commenced. This could therefore be due to a random effect of uncontrolled differences between housing or flocks. However, it is also possible that the pecking pans had a very rapid effect, shortening the birds’ beaks dramatically in just a few hours or days. In support, other authors [36] found that abrasive material in the feed trough worked immediately at the start of the experiment at 6 weeks of age. It was considered that this rapid effect could arise through a combination of the novelty of the abrasive materials and the softness of the beaks of young birds.

At 10–11 weeks, both mean side-beak length and mean top-beak length were significantly shorter in the birds provided with a pecking pan when birds of all beak status were considered together, and separately for both intact and IRBT birds. By 14–15 weeks, there was no significant difference in the two measurements when all birds were considered together, but the side-beak length of the intact birds remained shorter at this final visit when pans were provided. One explanation for the reduced effect of pans at the older age is that the birds’ beaks are continually growing and becoming stronger, so blunting may not have been achieved as easily as when the chicks were younger, having much softer beaks, or, more likely, blunting needs to occur continuously in growing birds and the use of the pans had substantially declined.

## 5. Conclusions

This study has highlighted factors that are likely to be important when considering a long-lasting enrichment to enhance bird welfare. It is the first to investigate, on multiple rearing farms, the potential benefit of blunting the bird’s beak naturally through normal levels of wear. This was achieved by provision of pecking pans containing an abrasive material and provided as an intended environmental enrichment. Evaluation of the results and methodology has given rise to questions to be addressed for future work based upon the improvements to the design and presentation of the pecking pan enrichment device and to some of the methods developed in this study. Despite some limitations in methodology, this study demonstrates that pecking pans for chicks/pullets may provide an environmental enrichment, with the potential added benefit of blunting the bird’s beak naturally through normal level of wear with a very quick onset of effect at 6–7 weeks and continuing to 10–11 weeks of age for all birds, and through to 14–15 weeks for birds with intact beaks. It remains to be determined whether the effect could be extended by presenting the abrasive material differently so that it remained clean and accessible. Furthermore, these preliminary results indicate genotypic variation in beak characteristics, which gives scope for selecting for genotypes with blunter, less damaging beaks.

## Figures and Tables

**Figure 1 animals-12-00674-f001:**
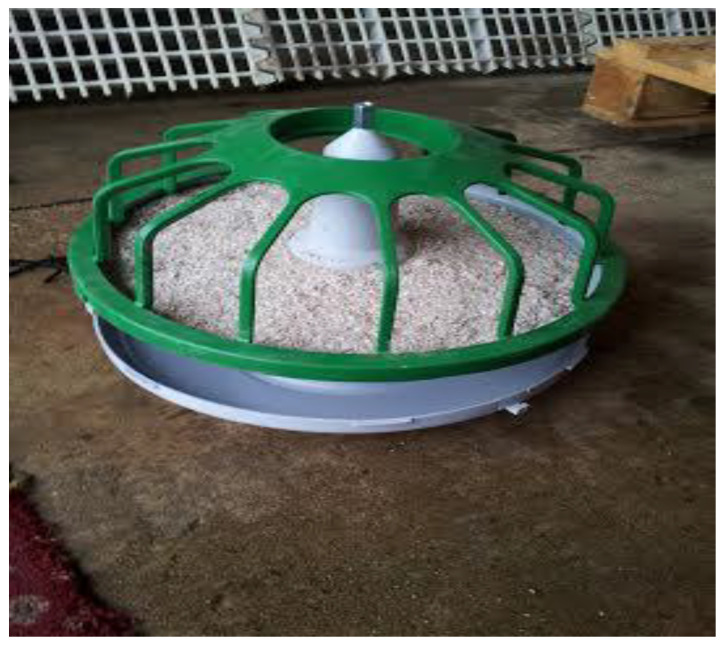
Photograph of the Vencomatic Pecking Pan.

**Figure 2 animals-12-00674-f002:**
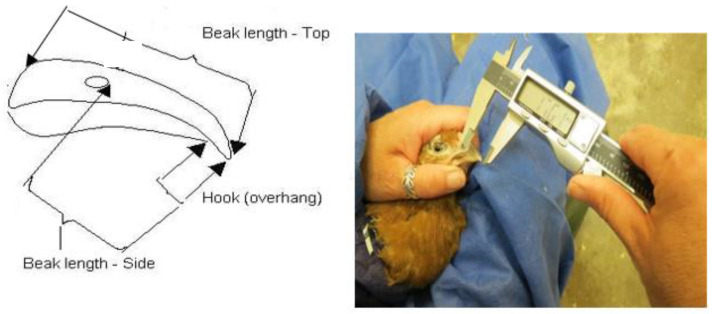
Beak length was measured with vernier callipers using methodology slightly adjusted from that of Fiks van Niekerk and Elson (2005).

**Figure 3 animals-12-00674-f003:**
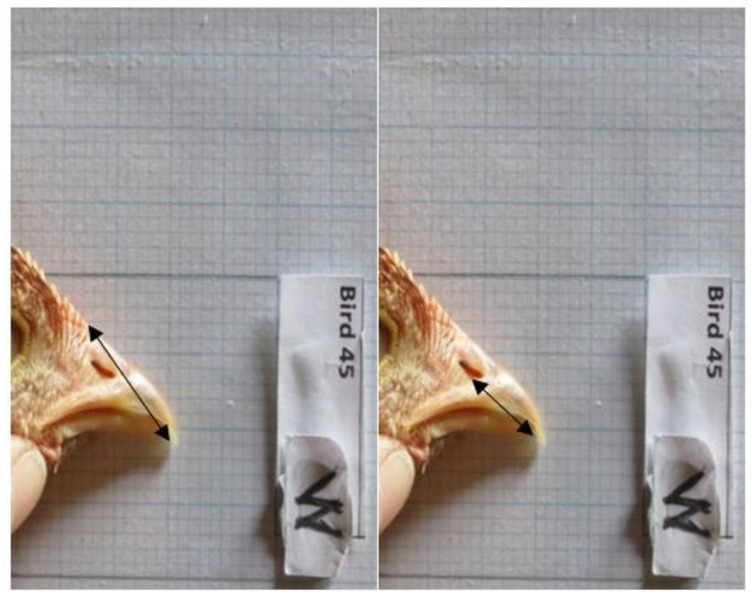
The black line on each image indicates the measurement that was taken on live birds, from left to right, top length and side length.

**Figure 4 animals-12-00674-f004:**
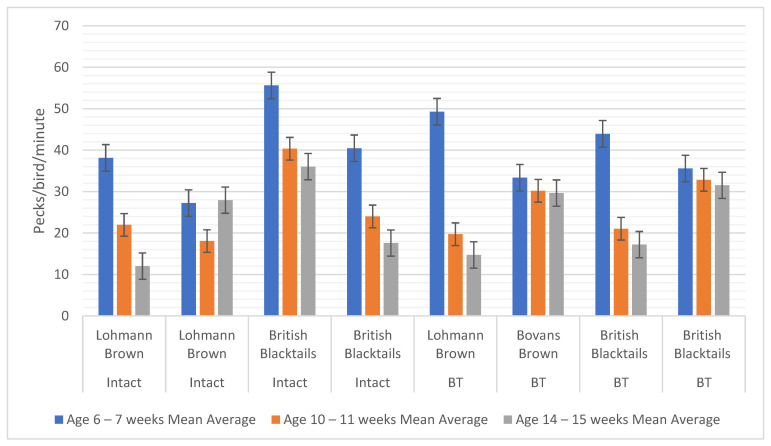
The decline with age in the pecking behaviour for eight flocks that were provided with a pecking pan (mean pecks/bird/min).

**Figure 5 animals-12-00674-f005:**
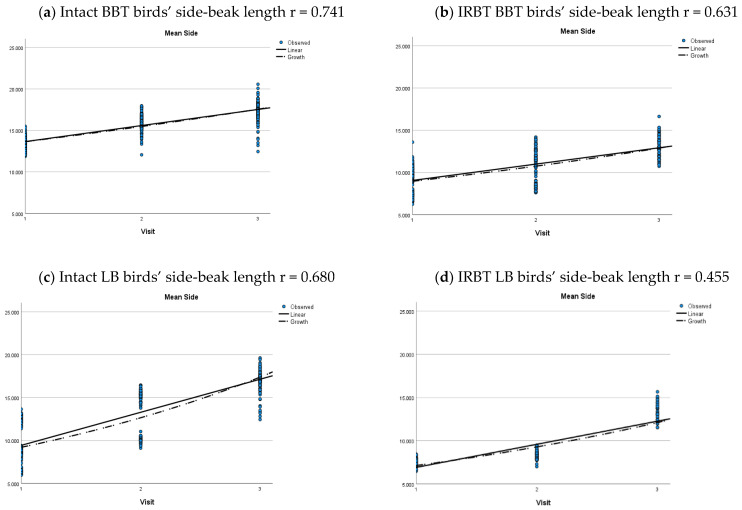
Growth curve of side-beak length for intact and IRBT flocks for each strain. (**a**) British blacktail birds with intact beaks; (**b**) British blacktail birds with trimmed beaks; (**c**) Lohmann Brown birds with intact beaks; (**d**) Lohmann Brown birds with trimmed beaks; (**e**) Bovan Brown birds with trimmed beaks. There were no Bovan Brown flocks with intact beaks in the study.

**Figure 6 animals-12-00674-f006:**
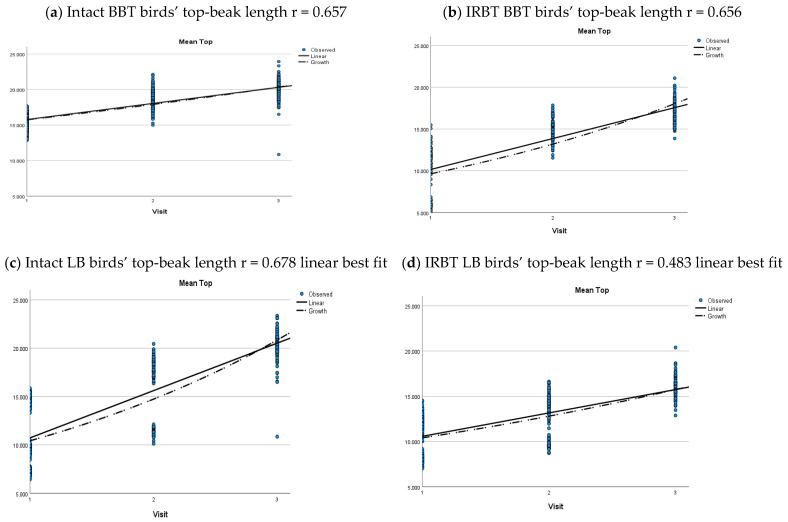
Growth curve of top-beak length for intact and IRBT flocks for each strain. (**a**) British blacktail birds with intact beaks; (**b**) British blacktail birds with trimmed beaks; (**c**) Lohmann Brown birds with intact beaks; (**d**) Lohmann Brown birds with trimmed beaks; (**e**) Bovan Brown birds with trimmed beaks. There were no Bovan Brown flocks with intact beaks in the study.

**Table 1 animals-12-00674-t001:** Flock information and pecking pan provision for all sixteen experimental groups.

Flock No.	Size	Genotype	Beak Status	Pecking Pans (1 Per 500 Birds) or Controls
1	10,000	British Blacktail	IRBT	No pecking pan
2	10,000	British Blacktail	Intact	No pecking pan
3	9000	Lohmann Brown	Intact	Pecking Pan
4	11,000	Lohmann Brown	IRBT	Pecking Pan
5	9000	Bovans Brown	IRBT	Pecking Pan
6	8000	Bovans Brown	IRBT	No pecking pan
7	3600	Lohmann Brown	Intact	No pecking pan
8	3600	British Blacktail	IRBT	Pecking Pan
9	10,000	Lohmann Brown	IRBT	No pecking pan
10	8000	British Blacktail	IRBT	Pecking Pan
11	6400	Lohmann Brown	Intact	Pecking Pan
12	6500	British Blacktail	Intact	Pecking Pan
13	4100	British Blacktail	Intact	No pecking pan
14	3400	Lohmann Brown	IRBT	No pecking pan
15	3300	British Blacktail	Intact	Pecking Pan
16	3600	British Blacktail	Intact	No Pecking Pan

**Table 2 animals-12-00674-t002:** Ethogram of feather pecking behaviour.

Behaviour	Definition
GFP	Soft gentle feather pecking, without pulling and removal of feathers
SFP	To peck or pull out the feathers of other birds with force
AP	Forceful pecking directed at the head and neck region
VP	Pecking directly at the vent area
CB	Pecking at exposed skin creating wounds, leading to cannibalistic pecking

**Table 3 animals-12-00674-t003:** Description of feather scoring.

0	Well feathered body part; feathers intact
1	Slight damage with feathers ruffled but body part completely or almost completely covered
2	Severe damage to feathers but localised naked areas (<5 cm^2^)
3	Severe damage to feathers and large naked areas (>5 cm^2^)
4	Severe damage to feathers, (>5 cm^2^) naked area and haemorrhage due to broken skin

**Table 4 animals-12-00674-t004:** Description of beak measurements see Figure 2 and Figure 3.

Beak Measurements	Description
Top of the beak	Top of the beak, at the frontal feather tract margin to tip
Right side of the beak	Nares to the tip of the beak

**Table 5 animals-12-00674-t005:** Means and confidence intervals of visual flock feather scores in intact and beak trimmed birds with and without access to pecking pans. There were no significant differences.

Body Region	Beak Status		Pecking Pan Provision	
	Intact	Trimmed	Pans	No Pans
Neck	0.40 (0.02–0.06)	0.03 (0.01–0.04)	0.02 (0.01–0.04)	0.04 (0.02–0.07)
Back	0.00 (0.00–0.03)	0.01 (0.01–0.02)	0.02 (0.01–0.02)	0.01 (0.00–0.02)
Tail	0.32 (0.00–0.03)	0.01 (0.27–0.38)	0.31 (0.27–0.35)	0.34 (0.27–0.40)

**Table 6 animals-12-00674-t006:** Flock mean feather scores (confidence intervals) in handled birds comparing beak status and pecking pan provision. There were no significant differences.

Body Region	Beak Status		Pecking Pan Provision	
	Intact	Trimmed	Pans	No Pans
Neck	0.10 (0.05–0.15)	0.06 (0.03–0.09)	0.09 (0.05–0.12)	0.07 (0.02–0.12)
Back	0.04 (0.00–0.00)	0.02 (0.01–0.00)	0.024 (0.00–0.04)	0.04 (0.00–0.88)
Tail	0.04 (0.37–0.62)	0.45 (0.32–0.59)	0.432 (0.32–0.55)	0.51 (0.38–0.66)
Wings	0.03 (0.00–0.06)	0.03 (0.00–0.74)	0.01 (0.04–0.03)	0.05 (0.01–0.09)

**Table 7 animals-12-00674-t007:** Median and inter-quartile range (IQR) of side-beak lengths (mm) for intact and IRBT flocks for all ages (6–7 weeks, 10–11 weeks, 14–15 weeks).

Age (Weeks)	Median (IQR) Side-Beak Length—Intact Flocks	Median (IQR) Side Beak-Length—IRBT Flocks	Mann–Whitney U	Standardized Test Statistic	*p*-Value
	*n* = 360	*n* = 330			
6–7	12.99 (4.38)	7.99 (2.66)	19,649	−15.23	<0.001
10–11	15.54 (1.25)	11.25 (3.76)	9263	−19.19	<0.001
14–15	17.32 (1.10)	13.42 (1.52)	2564	21.75	<0.001

**Table 8 animals-12-00674-t008:** Median and inter-quartile range (IQR) of top-beak lengths (mm) for intact and IRBT flocks for all ages (6–7 weeks, 10–11 weeks, 14–15 weeks).

Age (Weeks)	Median (IQR) Top-Beak Length—Intact Flocks	Median (IQR) Top Beak-Length—IRBT Flocks	Mann–Whitney U	Standardized Test Statistic	*p*-Value
	*n* = 360	*n* = 330			
6–7	14.73 (5.70)	7.99 (2.66)	19,649	−15.23	<0.001
10–11	18.33 (2.20)	13.72 (5.12)	10,391	−18.71	<0.001
14–15	20.23 (1.54)	16.48 (1.90)	3414	−21.43	<0.001

**Table 9 animals-12-00674-t009:** Median and inter-quartile range (IQR) of side-beak lengths (mm) for birds of all beak status, showing effect of pecking pan provision at (6–7 weeks, 10–11 weeks, 14–15 weeks).

Age (Weeks)	Median (IQR) Side-Beak Length with Pan	Median (IQR) Side Beak-Length with No Pan	Mann–Whitney U	Standardized Test Statistic	*p*-Value
	*n* = 360	*n* = 330			
6–7	9.12 (5.43)	10.13 (5.18)	50,838	−3.274	<0.001
10–11	12.04 (6.18)	13.97 (4.10)	46,046	−5.106	<0.001
14–15	15.16 (1.98)	15.21 (4.14)	57,659	−0.666	0.506

**Table 10 animals-12-00674-t010:** Median and inter-quartile range (IQR) of top-beak lengths (mm) for birds of all beak status, showing effect of pecking pan provision at (6–7 weeks, 10–11 weeks, 14–15 weeks).

Age (Weeks)	Median (IQR) for Top Length with Pan	Median (IQR) for Top Length with No Pan	Mann–Whitney U	Standardized Test Statistic	*p*-Value
	*n* = 360	*n* = 330			
6–7	10.20 (6.64)	12.22 (5.50)	53,608	−2.214	0.027
10–11	15.10 (7.40)	16.45 (4.47)	48,490	−4.171	<0.001
14–15	18.51 (3.78)	18.32 (3.87)	62,144	1.049	0.294

## Data Availability

Data on beak measurements are available at the Dryad open access repository: doi:10.5061/dryad.2bvq83bs3.

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
