# Peer review of "The Effect of Hard Pecking Enrichment during Rear on Feather Cover, Feather Pecking Behaviour and Beak Length in Beak-Trimmed and Intact-Beak Laying Hen Pullets"

_animals, 2022, doi:10.3390/ani12060674_

Round 1

Reviewer 1 Report

Animals-1594094 

“The effect of hard pecking enrichment during rear on feather cover, feather pecking behaviour and beak length in beak-trimmed and intact-beak laying hen pullet”

General Comments:

The authors should remove unsupported speculation statements and provide references that support their conclusion statements.

The authors use incorrect anatomical terminology.  The term “upper mandible” in not correct.

“The beak (rhamphotheca) consists of the maxilla (upper) and mandible (lower) jaw bones and their horny, keratinized sheaths (Fig. 2.9.)”

The authors provide conflicting measurement point in Figure 3 compared to Figure 2 and the text.  Figure 3 indicates that “top” beak length measurement (left) was above the nares at the top of the beak at the frontal feather tract margin and “side” beak length does not extend to the tip of the top beak but end at the end of the mandible (lower beak).

The authors have obvious errors in the titles of tables and column headings.

L307 Table 7 correct table title since this table included IRBT flocks not just “intact” flocks.

L309 Table 8 correct table title since this table included intact flocks not just “IRBT” flocks.

L309 Replace “side” with “top” in the title for beak length measurements.

Specific comments:

L9, L41 Provide a reference in the introduction that confirms that feather pulling is “painful” or delete this statement.

Reference [1] only mentions pain related to the practice of beak trimming not feather pulling. “However, beak trimming can cause acute and chronic pain and thus can be a serious welfare concern by itself (Petek and McKinstry, 2010; Gilani et al., 2013).

L16, L32L283 The claim on linearity in beak growth need to be confirmed by regression analysis, since the sample visit age is 6-7, 11-12, and 14-15 weeks a gap of 5 and 3.5 weeks which is not linear.

L17-18 This is speculation since you make no measure of beak “softness”.

L26 Since “trends” are not significant delete this statement from the abstract.

L58-59 Provide correct reference that beak trimming removes “up to one third of the upper and lower beak”. Replace reference 17 a review with the original research manuscript.

L97 Replace the phrase “mutilation of beak trimming” this wording is inflammatory hyperbole.

L103 Provide a reference that confirmed that “due to faecal contamination, pecking bouts significantly decreased over time”.

L122, L361 Use correct anatomical terminology.  The term “upper mandible” in not correct. Use maxilla.

The beak (rhamphotheca) consists of the maxilla (upper) and mandible (lower) jaw bones and their horny, keratinized sheaths (Fig. 2.9.)

The beak, or rhamphotheca, is a horny skin structure of keratinized epidermis covering the upper and lower jaws.

L150 Replace the term “breeds” with strains since none of these are recognized breeds.

https://poultrykeeper.com/chicken-breeds/

L154 Indicate the 2 flock that were raised on slats.

L233 and L236 Correct Figure 3 measurement points. The measurements in Figure 2 and Figure 3 do not match the text.  Figure 3 indicates that “top” beak length measurement (left) was above the nares at the top of the beak at the frontal feather tract margin and “side” beak length does not extend to the tip of the top beak but end at the end of the mandible (lower beak).

L307 Table 7 correct table title since this table included IRBT flocks not just “intact” flocks.

L309 Table 8 correct table title since this table included intact flocks not just “IRBT” flocks.

L309 Replace “side” with “top” in the title for beak length measurements.

L331, L406 Replace “visit” with flock ages.  State if the regressions were generated using visits (1,2,3) or ages (5-6, 11-121, 15-16 weeks of age. L259 states “visit” but age should be used.

L358 Correct reference citation since in reference [28] for all experiments 1-5 trials only lasted for 5 days after placement.

In all but experiment 1 reference 28 states chicks were placed at “1 day of age” and none lasted to day 10.

L358-362 Delete speculation on chick beak hardness.

L366-369 Delete speculation.  Standard poultry management requires cleaning feeding pan and removal of fecal contaminating daily.  Explain why you did not implement emptying and cleaning pans on a weekly schedule and also prevent perching access.

L371-373 Delete speculation.

L374 Provide correct reference.  Reference [18] is a review and the method was not IRBT but “ Beak trimming (also named beak tipping, beak mutilation, debeaking or partial beak amputation) is routine practice in the poultry industry. It typically involves the removal of 1/3 to 1/2 of the upper and lower mandibles using an electrically heated blade that both cuts and cauterizes the beak tissue. T

Reference [18] reports just the opposite of your statement. “They reported that neuromas were fund in the Brown egg layers whose beaks were trimmed at 5 wk of age or older but not in the young chickens that beak was trimmed at 1 day or 10 days of age (Gentle et al., 1997) or in young turkeys’ beak trimmed at 1 day of age (Gentle et al., 1995).”

L39 Replace reference [47] with a published manuscript.

Author Response

Dear Reviewer,

Thank you for your comments and suggestions. Please see word document attached.

Best wishes,

Paula Baker

Reviewer 2 Report

Authors aims to conduct a controlled investigation in commercial flocks on the effects of a hard pecking material on the behaviour and beak characteristics of beak-trimmed and intact-beak laying hen genotypes during rear.

Specific suggestions/comments are highlighted in the attached .pdf file.

The manuscript is very well written and show important results in terms of pecking behaviours in commercial laying hens, an important industry problem, with a multifactorial aetiology, but related with animal density, feeding, and environment enrichment. This manuscript recognize potential limitations on its design and the interpretation of its results.

Some minor specific comments:

- Please check for consistent formatting, as different font sizes and styles are seen in various places in the document.

- Check in table 10, title and body, if the medians are for top-beak or side-beak length in order to correct it.

- Please check for the reference formatting.

- Material and methods: Please include a statement about how groups were built, flocks were randomly allocated into pecking pan or no pecking pan or did authors use any other design.

Knowing this, it would be of great interest to evaluate if this happens also in backyard hens or fully free-ranging hens. Or the impact of flock size and animal density on pecking behavoiur.

Author Response

Dear Reviewer,

Thank you for your comments and suggestions. Please see word document attached for clarification and reponse.

Best wishes,

Paula Baker
